# Effectiveness of physical therapy for locomotive syndrome: A systematic review and meta-analysis protocol

Chadapa Rungruangbaiyok[1,2], Hiroyuki Ohtsuka[3,4,5]*, Charupa Lektip[1,2], Jiraphat Nawarat[1,2], Eiji Miyake[3,4], Keiichiro Aoki[3,6], Yasuko Inaba[3,4], Yoshinori Kagaya[3,4]

1 Department of Physical Therapy, School of Allied Health Sciences, Walailak University, Nakhon Si Thammarat, Thailand, 2 Movement Sciences and Exercise Research Center, Walailak University, Nakhon Si Thammarat, Thailand, 3 Graduate School of Health Sciences, Showa Medical University, Yokohama, Kanagawa, Japan, 4 Division of Physical Therapy, Department of Rehabilitation, School of Nursing and Rehabilitation Sciences, Showa Medical University, Yokohama, Kanagawa, Japan, 5 Institute of Clinical Epidemiology, Showa Medical University, Shinagawa, Tokyo, Japan, 6 Division of Occupational Therapy, Department of Rehabilitation, School of Nursing and Rehabilitation Sciences, Showa Medical University, Yokohama, Kanagawa, Japan

⊕ These authors contributed equally to this work.
* ohtsuka@nr.showa-u.ac.jp

## Abstract

Locomotive syndrome, characterized by impaired mobility owing to musculoskeletal disorders, poses a significant public health challenge, especially in aging populations. Locomotive syndrome limits physical activity, increases fall risk, leads to dependency, and diminishes quality of life. Effective interventions are urgently required. This systematic review and meta-analysis aims to evaluate the effectiveness of physical therapy in improving the symptoms of locomotive syndrome. A systematic evaluation of its effectiveness compared with other interventions is crucial for informing clinical practice and policy decisions. This systematic review and meta-analysis will follow the PRISMA-P guidelines. Studies involving individuals diagnosed with locomotive syndrome, without restrictions on age, sex, or location, will be included. The reviewed interventions will include exercise programs, manual therapy, neuromuscular stimulation, and balance training, which will be compared with the absence of interventions or alternative therapies. The primary outcomes will include improvements in functional mobility and physical performance, symptom reduction, and the progression of locomotive syndrome. Secondary outcomes will include adherence to therapy, safety, quality of life, and psychological well-being. Randomized controlled and non-randomized controlled trials published in peer-reviewed journals will be searched in PubMed, CENTRAL, CINAHL, PEDro, Ichushi Web, and the Thai-Journal Citation Index Center, without language restrictions. Independent reviewers will perform data extraction and assess the risk of bias. A meta-analysis will be conducted using RevMan 5.4 software, with subgroup analyses to address heterogeneity. The Grading

**Data availability statement:** No datasets were generated or analysed during the current study. All relevant data from this study will be made available upon study completion.

**Funding:** This study was supported by the School of Nursing and Rehabilitation Sciences Showa University Research Fund (Grant Numbers 2024No.4: H.O.). The funders had no role in study design, data collection and analysis, decision to publish, or preparation of the manuscript.

**Competing interests:** NO authors have competing interests.

of Recommendations Assessment, Development and Evaluation (GRADE) approach will be used to evaluate the certainty of evidence. This review aims to provide robust evidence on the effectiveness of physical therapy in managing locomotive syndrome and to potentially guide clinical practice and healthcare policy decisions.

## Introduction

Locomotive syndrome (LS), a condition characterized by impaired mobility owing to musculoskeletal disorders, has become a significant public health concern, particularly in the aging population [1–4]. This syndrome, which encompasses a range of conditions, such as osteoarthritis, sarcopenia, and osteoporosis, adversely affects quality of life by limiting physical activity and increasing the risk of falls and dependency [1,2,5,6].

The increasing prevalence of LS, particularly in aging populations, underscores the urgent need for effective interventions to mitigate its impact [1,2,5,7,8]. In Japan, where the concept of LS was first introduced, the aging population was particularly affected. The prevalence of LS among older adults is rising, with a corresponding increase in the risk of falls and fractures, which further exacerbates the associated disability and healthcare costs [1,2,4,5,7–10]. This trend is also not unique to Japan as other countries with aging populations are experiencing similar challenges [11–13].

Given these trends, effective interventions are urgently needed to prevent or slow LS progression. Physical therapy interventions, through a combination of strength training, balance exercises, gait training, pain management, and flexibility exercises, provide a comprehensive approach to managing LS. The robust evidence from several studies underscores the effectiveness of physical therapy in preventing or slowing the progression of this condition, thereby enhancing the quality of life and reducing healthcare costs associated with disability and dependency [10,14–24]. However, to ensure the optimal allocation of healthcare resources and the best outcomes for patients, it is critical to systematically evaluate the effectiveness of physical therapy compared with other available interventions.

Although physical therapy is commonly prescribed to address LS symptoms [2–4,18,19], its relative effectiveness compared to other treatments, such as pharmacological approaches, surgical options, or alternative therapies, is debatable [9,18,20,25–29]. LS encompasses distinct conditions, such as osteoarthritis, osteoporosis, and sarcopenia, which affect various organ systems, these conditions share common pathophysiological mechanisms that lead to impaired mobility and increased risk of falls. While previous reviews [30] have explored interventions for LS based on randomized controlled trials, their findings were limited by the small number of included studies, heterogeneity of interventions, and inconclusive results. Moreover, valuable insights from non-randomized trials were not included. To address these gaps, our review will synthesize evidence from both randomized and non-randomized studies, evaluate a broader range of physical therapy interventions, and assess clinically relevant outcomes, thus providing a more comprehensive understanding of LS management. Given that LS is characterized by overlapping musculoskeletal

conditions with shared mechanisms, grouping these conditions together allows for a holistic evaluation of interventions targeting overall locomotor function. Therefore, this systematic review and meta-analysis will aim to evaluate the effectiveness of physical therapy interventions in improving functional mobility and health-related quality of life in individuals with LS.

We hypothesize that physical therapy is more effective than other treatment modalities (e.g., pharmacological, surgical, or alternative therapies) in improving functional mobility and health-related quality of life in individuals with LS. The clinical relevance of physical therapy interventions will be assessed by measuring changes in the functional mobility and health-related quality of life that meet or exceed established minimal clinically important differences (MCIDs), thereby indicating a meaningful impact on patient function and daily activities.

This protocol outlines a systematic approach to synthesize the current evidence base and identify the clinical relevance of physical therapy interventions for LS across diverse populations.

## Methods

This systematic review protocol has been registered in the International Prospective Register of Systematic Reviews (PROSPERO; registration number CRD42024515983) and will be conducted and reported in accordance with the Preferred Reporting Items for Systematic Reviews and Meta-Analyses Protocols (PRISMA-P) statement [31]. The study has not yet commenced, and data collection will begin upon acceptance of this protocol and will follow the predefined search strategy. The entire data collection process is expected to be completed within 6 months. Details of the protocol structure are provided in S1 Appendix.

### Participants/population

Individuals diagnosed with LS [1–4] will be included, without restrictions on age, sex, or geographical location. Individuals who are not diagnosed with LS or those with other diagnoses will be excluded from the review. This approach will ensure a focused and comprehensive analysis of the population affected by LS. To confirm the diagnosis of LS in the included studies, we will require the use of validated diagnostic criteria or tools, such as the loco-check questionnaire, 25-question Geriatric Locomotive Function Scale (GLFS-25), stand-up test, or two-step test, as recommended by the Japanese Orthopaedic Association and related clinical guidelines [2,3,10].

### Interventions/exposures

This review will examine various physical therapy interventions in patients with LS. Interventions will include exercise programs, manual therapy, neuromuscular stimulation, and balance training.

### Comparators/controls

Comparators will consist of no intervention or alternative active interventions. This will enable comparisons across different therapeutic approaches.

### Main outcomes

The primary outcomes will include:

- Functional Mobility: Evaluated through measures such as walking speed, gait analysis, and daily activity performance [30,32,33].

- Physical Performance Measures: Assessed using tests of balance, strength, and endurance [19,30,32,33].

- Symptom Reduction: Defined as decreases in pain, stiffness, and other LS-related symptoms [5,9,30].

- Disease Progression: Monitored via changes in LS severity over time.

## Additional outcomes

The secondary outcomes will include:

- Adherence to Therapy: Defined as the extent to which patients adhere to their prescribed physical therapy regimens will be evaluated.

- Safety and Adverse Events: Documentation and analysis of any adverse events or safety concerns associated with the physical therapy interventions will be included.

- Quality of Life: Assessed using validated self-reported instruments such as SF-36 and EuroQoL 5-dimension, which have been employed in longitudinal studies assessing exercise habits and their impact on QOL in individuals with LS [34].

- Psychological Well-being: The impact of physical therapy on the psychological and emotional well-being of patients will be examined using validated instruments, including the psychological subitems of the GLFS-25 and, where available, the Geriatric Depression Scale (GDS-15) [2,35].

## Eligibility criteria

We will include randomized controlled trials (RCTs) and non-RCTs published in peer-reviewed journals, without restrictions on publication language. Review articles, conference abstracts, and letters to the editor will be excluded. These criteria will ensure that the review focuses on high-quality peer-reviewed studies that provide comprehensive data for rigorous analysis and synthesis while excluding sources that typically lack detailed data.

## Information sources and search strategy

The following databases will be used for the search: PubMed, Cochrane Central Register of Controlled Trials (CENTRAL), CINAHL, PEDro, Scopus, Ichushi Web (in Japanese), and Thai Journal Citation Index Center (in Thailand). The search will not be restricted by language; all available studies in these databases will be considered, focusing on human participants. Abstracts and conference proceedings will also be included, and the authors will be contacted for additional details, if necessary. Studies published in non-English journals will be translated before evaluation. Translations will be performed using a combination of professional translation tools (e.g., DeepL) and manual verification by researchers proficient in the respective languages to ensure accuracy. This comprehensive search strategy will ensure a thorough and inclusive collection of relevant studies for the systematic review. The primary search terms will include the following: "locomotive syndrome," "rehabilitation," "physical therapy," "exercise," "electrical stimulation," and "clinical trial." A preliminary version of the search strategy for the relevant databases will be included in the supplementary material (S2 Appendix).

## Data extraction (selection and coding)

The searches will be performed by two independent reviewers (HO and CR) using widely recognized databases. The search terms will include a combination of MeSH (Medical Subject Headings) terms from these databases and free-text search terms agreed upon by all authors. Rayyan will be used to manage the studies across different databases.

Title and abstract screening will be conducted independently by at least two reviewers selected from a team of seven (HO, CR, CL, JN, EM, YI, and YK). Studies that cannot be conclusively evaluated based on the title and abstract alone will undergo full-text review. In case of disagreement between the two reviewers, a third reviewer will be available for discussion to resolve the issue.

For data extraction, two independent reviewers (HO and CR) will collect detailed information on the study design and methodology, demographic and baseline characteristics of the participants, sample size, and measures of effect. Any discrepancies in judgment between the reviewers will be resolved through discussion with a third reviewer. Any missing data

in the articles will be requested from the study authors, as necessary. The data extraction process will be meticulously documented and organized using a standard Microsoft Excel spreadsheet.

## Risk of bias assessment

The risk of bias in the RCTs will be assessed using the Cochrane Risk of Bias tool (RoB 2.0). Two independent reviewers (HO and CR) will critically assess all the included studies. The evaluation will include the following items:

- Bias arising from the randomization process

- Bias due to deviations from the intended intervention

- Bias due to missing outcome data

- Bias in the measurement of outcome

- Bias in the selection of the reported result

- Overall effect

For each item, each study will be evaluated as having a low, unclear, or high risk of bias. Any discrepancies between the reviewers will be discussed and resolved by a third reviewer, if necessary.

The risk of bias in non-RCTs will be assessed using the Risk of Bias Assessment Tool for Nonrandomized Studies (RoBANS). Two independent reviewers (HO and CR) will critically assess all the included studies. The evaluation will include the following items:

- Selection of participants

- Confounding variables

- Measurement of exposure

- Blinding of the outcome assessment

- Incomplete outcome data

- Selective outcome reporting

For each item, each study will be evaluated as having a low, unclear, or high risk of bias. Any discrepancies between the reviewers will be discussed and resolved by a third reviewer, if necessary.

## Data synthesis

If numerous RCTs consistently corroborate their findings, a meta-analysis will be conducted. For this process, RevMan 5.4 software will be employed. For the analysis of continuous data, weighted mean differences (MD), including means and standard deviations, will be utilized. The standardized mean difference (SMD) will be applied to coalesce multiple measurements of identical outcome variables. To derive aggregate estimates, a random-effects model will be adopted, accompanied by forest plots to graphically represent the findings. The $I^2$ test will be used to assess heterogeneity; a value surpassing 50% in the $I^2$ test will indicate substantial heterogeneity, necessitating the execution of a subgroup analysis. Consequently, a comprehensive table summarizing the results will be compiled in alignment with the reported findings.

To assess publication bias, a funnel plot will be generated if 10 or more studies are included in a meta-analysis. If fewer than 10 studies are included, the Egger's regression test will be conducted to evaluate the potential for small-study effects.

To evaluate robustness of the meta-analysis findings, a sensitivity analysis will be performed by excluding studies assessed as having a high risk of bias or small sample sizes (e.g., < 30 participants per group). This analysis will help determine the extent to which these factors influence the overall results.

## Subgroup analysis

When multiple trials are present within each subgroup, the analyses will be stratified based on the participant demographics, including sex (male/female), nature of the intervention, and type of control group (either no intervention or active control). Furthermore, subgroup analyses will be conducted in cases with a significant degree of heterogeneity.

Additional subgroup factors will include age, disease severity, and disease duration, as these variables may significantly influence the responsiveness to physical therapy interventions. For instance, patients at early stages of the disease may respond differently to certain interventions compared with those in later stages. These analyses will help determine whether different demographic or clinical characteristics influence the effectiveness of physical therapy in managing LS, thereby providing more targeted and individualized treatment recommendations.

## Assessment of certainty of evidence

The certainty of evidence will be evaluated using the Grading of Recommendations, Assessment, Development, and Evaluations (GRADE) approach. This systematic method will be used to assess the quality of evidence across the domains of risk of bias, consistency of effects, imprecision, indirectness, and publication bias.

Each outcome will be rated as high, moderate, low, or very low. The initial rating for RCTs will be high; however, it might be downgraded based on the aforementioned domains. Conversely, the rating for observational studies will start as low but will be upgraded if the evidence shows a large effect, a dose-response gradient, or if all plausible biases reduce an apparent treatment effect.

Two independent reviewers (HO and CR) will conduct the GRADE assessment, and any discrepancies will be resolved through discussion or consultation with a third reviewer. The results of the GRADE assessment will be summarized in a table that will provide a clear and transparent evaluation of the certainty of the evidence for each outcome.

## Patient and public involvement

There will be no patient or public involvement in the design, conduct, reporting, or dissemination of this systematic review.

## Ethical considerations

No ethical approval will be required for this systematic review, as it will involve the analysis of data from previously published studies and will not involve direct contact with patients or the collection of primary data.

## Discussion

Several studies have explored the benefits of physical therapy for managing musculoskeletal conditions, thereby providing a foundation for our research. For instance, the meta-analysis by Liu and Latham (2009) highlighted the effectiveness of progressive resistance strength training in improving physical function in older adults [17]. Sherrington et al. (2011) conducted a meta-analysis demonstrating the benefits of exercise in preventing falls among older adults [18]. Furthermore, Fransen et al. (2015) reviewed the impact of exercise on knee osteoarthritis and found significant improvements in pain and physical function [19]. Despite these findings, the relative effectiveness of various physical therapy interventions, specifically for LS, remains unclear. Our research aims to fill this gap by systematically evaluating and synthesizing existing evidence and providing a comprehensive assessment of the impact of physical therapy on this condition.

Our systematic review and meta-analysis has several strengths, potentially enhancing the reliability and applicability of our findings. First, we will use a rigorous and transparent methodology adhering to the PRISMA-P guidelines, which will

ensure a systematic and unbiased approach to data collection, extraction, and analysis. The inclusion of both RCTs and non-RCTs will allow for a comprehensive evaluation of the evidence. Second, our extensive search strategy will include multiple databases and languages (English, Japanese, and Thai), ensuring a thorough and inclusive collection of relevant studies and minimizing the risk of publication bias. Third, we will employ robust statistical methods, such as random-effects models and subgroup analyses, to account for heterogeneity and provide a nuanced understanding of treatment effects across different populations and intervention types. Additionally, the use of Cochrane's risk-of-bias tool and the GRADE approach to assess the certainty of evidence will further strengthen the credibility of our findings.

Several practical and operational challenges are anticipated in conducting this study. Dealing with the heterogeneity of the included studies, such as variations in study design, intervention types, and outcome measures, should be carefully considered. This will be addressed through subgroup analyses and the use of random-effect models to account for variability. Efficient data extraction and management are critical when employing standardized forms and software, such as Rayyan, for study screening. Assessing the risk of bias and ensuring accurate data extraction will require multiple reviewers and a rigorous validation process.

When interpreting the results, it will be important to distinguish between statistically significant differences and clinically meaningful outcomes. Even if a result is statistically significant, it does not necessarily translate into a clinically relevant improvement. Therefore, we have considered MCIDs as a benchmark to determine whether the observed changes in functional mobility and quality of life are clinically significant. This ensures that the results not only demonstrate statistical validity but also have practical implications for patient care. For example, a small, statistically significant improvement in gait speed might not meet the threshold for MCIDs, meaning that while the intervention had an effect, it may not be sufficient to make a meaningful difference in a patient's daily life. This approach will allow clinicians to base treatment decisions on changes that are both statistically supported and clinically relevant, ensuring that the interventions offer real-world benefits to patients.

Adherence to physical therapy is another significant factor that can influence patient outcomes. We will evaluate the adherence rates and explore strategies to improve adherence, considering their feasibility and impact on study outcomes. In addition, understanding the long-term sustainability of physical therapies is crucial. We will, therefore, include follow-up studies to assess the duration of the observed effects and provide a comprehensive understanding of the long-term effectiveness of physical therapy.

This systematic review and meta-analysis can be subject to several limitations. First, the inclusion of both RCTs and non-RCTs may introduce variability in methodological quality, potentially influencing the pooled estimates. Second, considerable heterogeneity across interventions, study populations, and outcome measures may limit the comparability and generalizability of findings. While subgroup analyses and random-effects models will be used to address this, residual heterogeneity may persist. Third, some included studies may carry a high risk of bias, potentially weakening the overall strength of evidence. Fourth, the certainty of evidence, as assessed using the GRADE approach, may be low or very low for some outcomes, which should be considered when interpreting the results. Additionally, although we aim to assess clinical relevance using MCIDs, the definitions of these thresholds may vary across studies. Lastly, long-term outcomes and sustainability of physical therapy interventions remain underreported in the literature, which may limit our ability to draw conclusions about long-term effectiveness.

## Conclusion

This systematic review and meta-analysis aims to provide compelling evidence supporting the effectiveness of physical therapy for managing LS. By addressing the practical and operational challenges and employing a rigorous and comprehensive methodology, our research will contribute to the optimization of physical therapy interventions for this condition. These findings can guide clinical practice and assist healthcare providers in making evidence-based decisions, ultimately enhancing the quality of life of individuals with LS.

## Supporting information

**S1 Appendix. Detailed structure of the systematic review protocol, including objectives, eligibility criteria, and planned outcomes.**
(DOCX)

**S2 Appendix. Preliminary search strategies for all databases to be used in the review, including search terms and syntax examples.**
(DOCX)

## Acknowledgments

We thank Ms. Tomoko Morimasa and Ms. Asae Ito, (librarian, Showa University) for their advice on creating the search strategy. We would also like to thank Editage (www.editage.com) for the English language editing.

## Author contributions

**Conceptualization:** Chadapa Rungruangbaiyok, Hiroyuki Ohtsuka, Yoshinori Kagaya.

**Data curation:** Chadapa Rungruangbaiyok, Hiroyuki Ohtsuka.

**Formal analysis:** Chadapa Rungruangbaiyok, Hiroyuki Ohtsuka.

**Funding acquisition:** Chadapa Rungruangbaiyok, Hiroyuki Ohtsuka, Eiji Miyake, Keiichiro Aoki, Yasuko Inaba, Yoshinori Kagaya.

**Investigation:** Chadapa Rungruangbaiyok, Hiroyuki Ohtsuka.

**Methodology:** Hiroyuki Ohtsuka.

**Project administration:** Chadapa Rungruangbaiyok, Hiroyuki Ohtsuka, Yoshinori Kagaya.

**Resources:** Hiroyuki Ohtsuka.

**Software:** Chadapa Rungruangbaiyok, Hiroyuki Ohtsuka.

**Supervision:** Chadapa Rungruangbaiyok, Hiroyuki Ohtsuka, Charupa Lektip, Jiraphat Nawarat, Eiji Miyake, Keiichiro Aoki, Yasuko Inaba, Yoshinori Kagaya.

**Validation:** Chadapa Rungruangbaiyok, Hiroyuki Ohtsuka, Charupa Lektip, Jiraphat Nawarat, Eiji Miyake, Keiichiro Aoki, Yasuko Inaba, Yoshinori Kagaya.

**Visualization:** Chadapa Rungruangbaiyok, Hiroyuki Ohtsuka, Charupa Lektip, Jiraphat Nawarat, Eiji Miyake, Keiichiro Aoki, Yasuko Inaba, Yoshinori Kagaya.

**Writing – original draft:** Chadapa Rungruangbaiyok, Hiroyuki Ohtsuka.

**Writing – review & editing:** Chadapa Rungruangbaiyok, Hiroyuki Ohtsuka, Charupa Lektip, Jiraphat Nawarat, Eiji Miyake, Keiichiro Aoki, Yasuko Inaba, Yoshinori Kagaya.

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
