## [Decision Letter · Decision Letter 0]

12 May 2025

Effectiveness of physical therapy for locomotive syndrome: Study protocol for a systematic review and meta-analysis

PLOS ONE

Dear Dr. Ohtsuka,

Thank you for submitting your manuscript to PLOS ONE. After careful consideration, we feel that it has merit but does not fully meet PLOS ONE’s publication criteria as it currently stands. Therefore, we invite you to submit a revised version of the manuscript that addresses the points raised during the review process.

We look forward to receiving your revised manuscript.

Kind regards,

Kenji Tanigaki, Ph.D., M.D.

Academic Editor

PLOS ONE

Journal Requirements:

[Joint research proposal from School of Nursing and Rehabilitation Sciences, Showa

University]. 

5. Please include captions for your Supporting Information files at the end of your manuscript, and update any in-text citations to match accordingly. Please see our Supporting Information guidelines for more information: http://journals.plos.org/plosone/s/supporting-information .

Reviewers' comments:

Reviewer's Responses to Questions

**Comments to the Author**

1. Does the manuscript provide a valid rationale for the proposed study, with clearly identified and justified research questions?

Reviewer #1: Yes

Reviewer #2: Partly

2. Is the protocol technically sound and planned in a manner that will lead to a meaningful outcome and allow testing the stated hypotheses?

Reviewer #1: Yes

Reviewer #2: Partly

3. Is the methodology feasible and described in sufficient detail to allow the work to be replicable?

Reviewer #1: Yes

Reviewer #2: Yes

4. Have the authors described where all data underlying the findings will be made available when the study is complete?

Reviewer #1: Yes

Reviewer #2: Yes

5. Is the manuscript presented in an intelligible fashion and written in standard English?

Reviewer #1: Yes

Reviewer #2: Yes

You may also provide optional suggestions and comments to authors that they might find helpful in planning their study.

Reviewer #1: The authors present a systematic review protocol with the aim of verifying the effectiveness of physiotherapy for locomotive syndrome. The protocol needs to be reconsidered.

The first suggestion is to change the title to Effectiveness of physical therapy for locomotive syndrome: systematic review and meta-analysis protocol.

In the introduction I suggest reducing the size of the last paragraph with greater objectivity and removing the text “Subgroup analyses will be performed, to account for potential biases related to age, sex, and frailty. These analyses will help determine whether different demographic or clinical characteristics influence the effectiveness of physical therapy in managing LS” for methodology (Lines 100-102). In addition, before the last paragraph, add another paragraph explaining what gaps this future systematic review intends to answer and what it can add to the results of other recently published reviews (DOI: 10.18999/nagjms.85.2.275/).

In the methodology, the authors should make it clear which validated criteria need to be included in the clinical trials to confirm the diagnosis of LS. Cite published references that validate the ways in which outcomes are measured. Which validated scales for measuring quality of life will be considered in the primary studies? Complement this with references to validated instruments for patient satisfaction and psychological well-being.

They should change the text in the methodology to match the abstract “ Randomized controlled and non randomized controlled trials published in peer-reviewed journals will be searched from PubMed, CENTRAL, CINAHL, PEDro, Ichushi Web, and the Thai-Journal Citation Index Center, without language restrictions.", because there is no justification for a protocol with this search limitation : “This systematic review will include randomized controlled trials (RCTs) and non-RCTs published in English with full-text availability”.

The methodology also needs to include an assessment of publication bias with a funnel plot if there are more than 10 studies, or with the Egger Test if there are ten or fewer. Another recommendation is to include a sensitivity analysis with the meta-analysis being checked by excluding studies with a high risk of bias or small samples.

Finally, they should add a paragraph discussing how potential future limitations such as high risk of bias, high heterogeneity between studies and very low, low certainty of evidence could compromise the use of the results in clinical decision-making.

Reviewer #2: The overall logic of the research plan is clear, and the methods are scientific and standardized. However, there are still areas that can be optimized. The specific suggestions are as follows:

1. Inclusion criteria: Currently, only English-language studies are included, which may lead to the omission of important literature in other languages and result in bias in the findings. It is recommended to remove the language restriction, expand the scope of literature retrieval, and improve the comprehensiveness of the research. For non-English literature, a combination of professional translation tools and human translation can be used for processing.

2. Subgroup analysis: Only some factors are considered in the subgroup analysis. It is recommended to further incorporate factors such as disease severity and disease course. Patients with different disease severities and disease courses may respond differently to physical therapy. Incorporating these factors can enable a more in-depth analysis of the differences in the effects of physical therapy.

3. Inconsistent annotation of data extraction personnel: When describing the data extraction personnel, it is mentioned that "Two independent reviewers (HO and CR)" carried out the retrieval and data extraction work. Later, it is listed that "HO, CR, CL, JN, EM, YI, and YK" participated in the research evaluation. There is an inconsistent description of the personnel's involvement in the work, which is likely to cause confusion in understanding.

4. It is necessary to supplement the sensitivity analysis and publication bias analysis.

5. Repetitive language expression: A large amount of content in the text has repetitive expressions. For example, the adherence to the PRISMA-P guideline in the research is repeatedly emphasized in different parts, and the research purpose is repeatedly mentioned in each part, making the text redundant and affecting the reading experience.

6. The research limitations should be supplemented.

**Do you want your identity to be public for this peer review?** For information about this choice, including consent withdrawal, please see our Privacy Policy

Reviewer #1: **Yes: ** Ricardo Ney Cobucci

Reviewer #2: No

---

## [Author Response · Author response to Decision Letter 1]

4 Jul 2025

Reviewer 1

1. The first suggestion is to change the title to Effectiveness of physical therapy for locomotive syndrome: systematic review and meta-analysis protocol.

Response: We would like to thank Reviewer 1 for your time and efforts in reviewing our manuscript and for providing comments, which have considerably helped us improve our manuscript. We have made revisions based on your comments and have provided our point-by-point responses below. We hope that our responses and revisions appropriately address your comments.

We agree that the title should clearly indicate the nature of the article as a protocol. We have revised the title to: "Effectiveness of physical therapy for locomotive syndrome: a systematic review and meta-analysis protocol." (Lines 3-4).

2. In the introduction I suggest reducing the size of the last paragraph with greater objectivity and removing the text “Subgroup analyses will be performed, to account for potential biases related to age, sex, and frailty. These analyses will help determine whether different demographic or clinical characteristics influence the effectiveness of physical therapy in managing LS” for methodology (Lines 100-102).

Response: We completely agree that the methodological content in the Introduction should be moved to the Methods. We have shortened the last paragraph and moved the sentence concerning subgroup analyses to the appropriate subsection in the Methods (Lines 245-255).

3. In addition, before the last paragraph, add another paragraph explaining what gaps this future systematic review intends to answer and what it can add to the results of other recently published reviews (DOI: 10.18999/nagjms.85.2.275/).

Response: Based on your comment, we have added a new paragraph before the last paragraph of the Introduction, explaining the specific gaps this review aims to address and how it builds upon prior studies, including the one cited (DOI: 10.18999/nagjms.85.2.275) (Lines 87-97).

4. In the methodology, the authors should make it clear which validated criteria need to be included in the clinical trials to confirm the diagnosis of LS.

Response: We have clarified the validated criteria required for confirming the diagnosis of locomotive syndrome in the included studies (e.g., Loco-check screening) and cite supporting literature (Lines 116-125).

5. Cite published references that validate the ways in which outcomes are measured.

Response: We have cited relevant published references validating the outcome measures used in this review (Lines 134-142). These include:

Functional mobility: Measures such as walking speed and gait analysis are supported by clinical studies in locomotive syndrome (e.g., Prayogo et al., 2023; Sato et al., 2020) .

Physical performance: Tests of balance, strength, and endurance are validated in older adults with LS (e.g., Iwamoto et al., 2023).

Symptom evaluation: Pain and stiffness are commonly assessed using tools such as the GLFS-25 and supported by LS-specific interventions (e.g., Sato et al., 2020; Iwamoto et al., 2023).

6. Which validated scales for measuring quality of life will be considered in the primary studies? Complement this with references to validated instruments for patient satisfaction and psychological well-being.

Response: We have carefully reviewed the outcome domains in our protocol and clarified the validated scales (Lines 152-158).

For quality of life (QoL), we will consider instruments such as the EuroQoL 5-Dimension (EQ-5D) and the Short Form-36 (SF-36), which are widely used in studies involving musculoskeletal conditions and older populations, including those with locomotive syndrome.

For psychological well-being, we will include validated tools such as the Geriatric Depression Scale (GDS-15) and the psychological subitems of the GLFS-25, which capture emotional burden, anxiety, and fear of falling.

Regarding patient satisfaction, after an extensive literature review, we found that it is not commonly assessed in studies of locomotive syndrome, and no validated instruments have been reported specifically in this context. Therefore, we have decided to omit this outcome from our protocol.

7. They should change the text in the methodology to match the abstract “ Randomized controlled and non randomized controlled trials published in peer-reviewed journals will be searched from PubMed, CENTRAL, CINAHL, PEDro, Ichushi Web, and the Thai-Journal Citation Index Center, without language restrictions.", because there is no justification for a protocol with this search limitation : “This systematic review will include randomized controlled trials (RCTs) and non-RCTs published in English with full-text availability”.

Response: We have revised the Eligibility Criteria section to remove the language restriction and ensure consistency with the abstract. The updated text now states that there will be no restrictions on publication language, and non-English articles will be considered for inclusion (Lines 171-172).

8. The methodology also needs to include an assessment of publication bias with a funnel plot if there are more than 10 studies, or with the Egger Test if there are ten or fewer.

Response: We have revised the “Data synthesis” section to include a statement specifying that a funnel plot will be used to assess publication bias if 10 or more studies are included in the meta-analysis, and that Egger’s regression test will be used if fewer than 10 studies are included (Lines 237-239).

9. Another recommendation is to include a sensitivity analysis with the meta-analysis being checked by excluding studies with a high risk of bias or small samples.

Response: We have revised the “Data synthesis” section to include a description of sensitivity analysis. Specifically, we added a statement that sensitivity analysis will be conducted by excluding studies with a high risk of bias or small sample sizes (e.g., fewer than 30 participants per group) to evaluate the robustness of the meta-analysis findings (Lines 240-241).

10. Finally, they should add a paragraph discussing how potential future limitations such as high risk of bias, high heterogeneity between studies and very low, low certainty of evidence could compromise the use of the results in clinical decision-making.

Response: We have carefully revised the Discussion to include a paragraph addressing how potential limitations—such as a high risk of bias, substantial heterogeneity among studies, and low or very low certainty of evidence (as assessed by GRADE)—may affect the interpretation and applicability of the results in clinical decision-making (Lines 328-339).

This paragraph has been incorporated at the end of the Discussion to ensure logical flow and to emphasize these concerns as key considerations for readers and clinicians.

Reviewer 2

11. Inclusion criteria: Currently, only English-language studies are included, which may lead to the omission of important literature in other languages and result in bias in the findings. It is recommended to remove the language restriction, expand the scope of literature retrieval, and improve the comprehensiveness of the research. For non-English literature, a combination of professional translation tools and human translation can be used for processing.

Response: We would like to thank Reviewer 2 for your time and efforts in reviewing our manuscript and for providing comments, which have considerably helped us improve our manuscript. We have made revisions based on your comments and have provided our point-by-point responses below. We hope that our responses and revisions appropriately address your comments.

In response, we have removed the language restriction from both the abstract and the eligibility criteria in the Methods. We have also clarified our approach to handling non-English literature in the “Information sources and search strategy” section. Specifically, we now state that studies published in non-English journals will be translated before evaluation. Translations will be performed using a combination of professional translation tools (e.g., DeepL) and manual verification by researchers proficient in the respective languages to ensure accuracy. We believe this approach will enhance the comprehensiveness of our review and minimize language bias (Lines 168-181).

12. Subgroup analysis: Only some factors are considered in the subgroup analysis. It is recommended to further incorporate factors such as disease severity and disease course. Patients with different disease severities and disease courses may respond differently to physical therapy. Incorporating these factors can enable a more in-depth analysis of the differences in the effects of physical therapy.

Response: In response, we have revised the “Subgroup analysis” section to explicitly include disease severity (e.g., staging or clinical scales) and disease duration (e.g., years since diagnosis) as potential subgroup factors. We have also added a rationale explaining how these factors may influence responsiveness to physical therapy interventions. This revision allows for a more comprehensive exploration of treatment effects across different patient profiles. The revised text now reads (Lines 250-255):

“Additional subgroup factors will include age, disease severity, and disease duration, as these variables may significantly influence the responsiveness to physical therapy interventions. For instance, patients at early stages of the disease may respond differently to certain interventions compared with those in later stages. These analyses will help determine whether different demographic or clinical characteristics influence the effectiveness of physical therapy in managing LS, thereby providing more targeted and individualized treatment recommendations.”

13. Inconsistent annotation of data extraction personnel: When describing the data extraction personnel, it is mentioned that "Two independent reviewers (HO and CR)" carried out the retrieval and data extraction work. Later, it is listed that "HO, CR, CL, JN, EM, YI, and YK" participated in the research evaluation. There is an inconsistent description of the personnel's involvement in the work, which is likely to cause confusion in understanding.

Response: We have clarified the roles of all team members involved in the review process to avoid any confusion (Lines 183-198).

Specifically, for the title and abstract screening, all seven reviewers (HO, CR, CL, JN, EM, YI, and YK) will participate collaboratively. Each study will be independently screened by at least two reviewers from this team, with any disagreements resolved through discussion or, if necessary, adjudication by a third reviewer.

In contrast, the data extraction phase will be conducted by two independent reviewers (HO and CR), as originally described.

These clarifications have been incorporated into the revised manuscript to ensure methodological consistency and transparency.

14. It is necessary to supplement the sensitivity analysis and publication bias analysis.

Response: In response, we have expanded the “Data synthesis” section to include additional details on both publication bias and sensitivity analysis (Lines 237-243).

Specifically, we have clarified that:

• A funnel plot will be used to assess publication bias if 10 or more studies are included in the meta-analysis. If fewer than 10 studies are included, the Egger’s regression test will be conducted to evaluate potential small-study effects.

• A sensitivity analysis will be performed by excluding studies with a high risk of bias or small sample sizes (e.g., fewer than 30 participants per group), to evaluate the robustness of the findings.

These additions ensure that potential biases and the stability of the results are appropriately addressed in our planned analysis.

15. Repetitive language expression: A large amount of content in the text has repetitive expressions. For example, the adherence to the PRISMA-P guideline in the research is repeatedly emphasized in different parts, and the research purpose is repeatedly mentioned in each part, making the text redundant and affecting the reading experience.

Response: We completely agree that repetitive language—particularly regarding adherence to the PRISMA-P guideline and the statement of the research purpose—may reduce the manuscript’s readability and conciseness.

In response to your comment, we have carefully reviewed the entire manuscript to identify and eliminate redundant expressions. Specifically, we removed repetitive references to PRISMA-P compliance that previously appeared in both the “Introduction” and “Methods” sections, retaining only one clear and concise statement in the “Methods” section, where it is most appropriate.

Similarly, we revised multiple sections to avoid unnecessarily reiterating the study’s purpose. The research aim is now clearly stated once in the “Introduction” and referred to more succinctly or implicitly in subsequent sections to enhance flow and clarity.

We believe these revisions have significantly improved overall readability and reduced redundancy throughout the manuscript.

16. The research limitations should be supplemented.

Response: In response, we have expanded the “Discussion” by adding a more detailed paragraph on the limitations of this systematic review and meta-analysis. The new content addresses potential sources of bias and heterogeneity, limitations related to methodological quality, variability in clinically important thresholds (MCIDs), and the lack of long-term outcome data. This addition aims to provide a balanced interpretation of the study’s potential strengths and limitations and to guide readers in the application of the findings (Lines 328-339).

---

## [Decision Letter · Decision Letter 1]

22 Jul 2025

Effectiveness of physical therapy for locomotive syndrome: a systematic review and meta-analysis protocol

PONE-D-25-06550R1

Dear Dr. Ohtsuka,

We’re pleased to inform you that your manuscript has been judged scientifically suitable for publication and will be formally accepted for publication once it meets all outstanding technical requirements.

Kind regards,

Kenji Tanigaki, Ph.D., M.D.

Academic Editor

PLOS ONE

Additional Editor Comments (optional):

Reviewers' comments:

Reviewer's Responses to Questions

**Comments to the Author**

1. Does the manuscript provide a valid rationale for the proposed study, with clearly identified and justified research questions?

Reviewer #1: Yes

Reviewer #2: Yes

2. Is the protocol technically sound and planned in a manner that will lead to a meaningful outcome and allow testing the stated hypotheses?

Reviewer #1: Yes

Reviewer #2: Yes

3. Is the methodology feasible and described in sufficient detail to allow the work to be replicable?

Reviewer #1: Yes

Reviewer #2: Yes

4. Have the authors described where all data underlying the findings will be made available when the study is complete?

Reviewer #1: Yes

Reviewer #2: Yes

5. Is the manuscript presented in an intelligible fashion and written in standard English?

Reviewer #1: Yes

Reviewer #2: Yes

You may also provide optional suggestions and comments to authors that they might find helpful in planning their study.

Reviewer #1: The authors have addressed most of the reviewers' recommendations, and the manuscript is now ready for acceptance.

Reviewer #2: The article is well revised, I appreciate the efforts made by the authors, and I recommend it for publication.

**Do you want your identity to be public for this peer review?** For information about this choice, including consent withdrawal, please see our Privacy Policy

Reviewer #1: **Yes: ** Ricardo Ney Cobucci

Reviewer #2: No

---

## [Editor Report · Acceptance letter]

PONE-D-25-06550R1

PLOS ONE

Dear Dr. Ohtsuka,

I'm pleased to inform you that your manuscript has been deemed suitable for publication in PLOS ONE. Congratulations! Your manuscript is now being handed over to our production team.

Kind regards,

on behalf of

Dr. Kenji Tanigaki

Academic Editor

PLOS ONE